# CT in relation to RT-PCR in diagnosing COVID-19 in The Netherlands: A prospective study

Hester A. Gietema[1,2], Noortje Zelis[3,4], J. Martijn Nobel[1,5], Lars J. G. Lambriks[3], Lieke B. van Alphen[6,7], Astrid M. L. Oude Lashof[6,8], Joachim E. Wildberger[1,4], Irene C. Nelissen[3], Patricia M. Stassen[3,7] *

1 Department of Radiology and Nuclear Medicine, Maastricht University Medical Centre, Maastricht, The Netherlands, 2 GROW School for Oncology and Developmental Biology, Maastricht University, Maastricht, The Netherlands, 3 Department of Internal Medicine, Division of General Internal Medicine, Section Acute Medicine, Maastricht University Medical Centre, Maastricht, The Netherlands, 4 CARIM School for Cardiovascular Diseases, Maastricht University, Maastricht, The Netherlands, 5 School of Health Professions Education, Maastricht University, Maastricht, The Netherlands, 6 Department of Medical Microbiology, Maastricht University Medical Centre, Maastricht, The Netherlands, 7 School CAPHRI, Care and Public Health Research Institute, Maastricht University, Maastricht, The Netherlands, 8 School of Nutrition and Translational Research in Metabolism, Maastricht University, Maastricht, The Netherlands

* p.stassen@mumc.nl

**Data Availability Statement:** All relevant data are available within the paper and its Supporting Information files.

## Abstract

### Introduction

Early differentiation between emergency department (ED) patients with and without corona virus disease (COVID-19) is very important. Chest CT scan may be helpful in early diagnosing of COVID-19. We investigated the diagnostic accuracy of CT using RT-PCR for SARS-CoV-2 as reference standard and investigated reasons for discordant results between the two tests.

### Methods

In this prospective single centre study in the Netherlands, all adult symptomatic ED patients had both a CT scan and a RT-PCR upon arrival at the ED. CT results were compared with PCR test(s). Diagnostic accuracy was calculated. Discordant results were investigated using discharge diagnoses.

### Results

Between March 13th and March 24th 2020, 193 symptomatic ED patients were included. In total, 43.0% of patients had a positive PCR and 56.5% a positive CT, resulting in a sensitivity of 89.2%, specificity 68.2%, likelihood ratio (LR)+ 2.81 and LR- 0.16. Sensitivity was higher in patients with high risk pneumonia (CURB-65 score ≥3; n = 17, 100%) and with sepsis (SOFA score ≥2; n = 137, 95.5%). Of the 35 patients (31.8%) with a suspicious CT and a negative RT-PCR, 9 had another respiratory viral pathogen, and in 7 patients, COVID-19 was considered likely. One of nine patients with a non-suspicious CT and a positive PCR had developed symptoms within 48 hours before scanning.

**Funding:** The authors received no specific funding for this work.

**Competing interests:** The authors have declared that no competing interests exist.

## Discussion

The accuracy of chest CT in symptomatic ED patients is high, but used as a single diagnostic test, CT can not safely diagnose or exclude COVID-19. However, CT can be used as a quick tool to categorize patients into "probably positive" and "probably negative" cohorts.

## Introduction

Corona virus disease 2019 (COVID-19) [1–3] is a highly contagious disease caused by Severe Acute Respiratory Syndrome Coronavirus 2 (SARS-CoV-2). Early differentiation between patients with and without the disease is extremely important [1–3], particularly in patients who visit the emergency department (ED) and patients who are admitted to the hospital. This differentiation is necessary to be able to select patients who need to be isolated to protect other patients and health care personnel [4].

Currently, reverse-transcriptase-polymerase-chain-reaction (RT-PCR) is the reference standard in diagnosing COVID-19. However, RT-PCR may have suboptimal sensitivity, for instance because in early stages of COVID-19, the viral load is below detection limit or because of technical issues, i.e. sampling errors [5]. In addition, in practice, it may take up to 24 hours to get a test result [6], although same day results are achieved most of the times.

As most COVID-19 patients present with pneumonia, computed tomography (CT) scanning of the thorax could be helpful in screening and diagnosing. In addition, CT has the advantage that the results can be available almost directly [7, 8]. Chest CT can show characteristic findings including areas of ground-glass, with or without signs of reticulation (so called "crazy paving pattern"), consolidative pulmonary opacities in advanced stages and the "reverse halo" sign [3, 7–12]. Since peripheral areas of ground glass are a hallmark of early COVID-19, which can easily be missed at chest X-rays, CT scanning has an advantage over chest X-rays in the early stages of COVID-19 [13]. Because COVID-19 related changes can indeed be found on CT scans, some studies suggest that CT scanning could be helpful in discriminating between COVID-19 positive and COVID-19 negative patients at the ED [8, 13–16]. The value of CT is, however, debated because of suspected lack of discriminatory value [17–21]. Complicating this debate, is that the sensitivity of PCR may be suboptimal, which makes it difficult to compare the two tests [5]. Furthermore, the added value of a diagnostic test is effected by the prevalence of disease and data on the performance of CT in a population with a moderate prevalence of COVID-19 are scarce.

In this prospective study, we therefore investigated the diagnostic accuracy of CT scanning in detecting COVID-19 in a population with suspected COVID-19 presenting at the ED using (repeated) RT-PCR testing as reference standard. Since the reference standard has been reported to be suboptimal, we further investigated reasons for discordant test results.

## Methods

### Setting

This study was performed in Maastricht University Medical Centre (MUMC+), a secondary and tertiary care hospital with around 700 beds and 23,000 ED visits a year. The study was approved by the medical ethics committee of MUMC+ (METC 2020–1564) and informed consent was waived. This study is reported in line with the STARD guidelines for diagnostic accuracy studies [22].

## Patients and design

All adult (18 years or older) patients who consecutively visited the ED between March 13 and March 24 2020 with respiratory symptoms including dyspnoea, coughing, sore throat and fever were scanned in a mobile CT scan unit. Of these patients, a nasopharyngeal and/or oropharyngeal swab was taken and tested for presence of SARS-CoV-2. If the first PCR was negative, a second PCR was performed within 48 hours after the first test in patients who were still admitted to the hospital, if deemed indicated by the clinicians, e.g. if no alternative diagnosis was made. We included all symptomatic patients who received a chest CT and at least one PCR test for detection of COVID-19. Five patients who were immediately intubated upon arrival at the ED received no CT scanning and were therefore not included in our study.

## RT-PCR

Laboratory confirmation of SARS-CoV-2 was performed with RT-PCR assay. First, RNA was extracted from clinical samples with the MagNA Pure 96 system (Roche, Germany). RT-PCR assay was performed using as targets RdrP-gene and E-gene according to the previous published protocol. Thermal cycling was performed on an Quantstudio 5 (Applied Biosystems, US). Validation of the method was performed in accordance with the protocol established by the RIVM and ErasmusMC [23]. Oligonucleotides were synthesised and provided by Tib-Molbiol (Berlin, Germany) or by Biolegio (Netherlands). All swabs were also tested for Influenza A, B, respiratory syncytial virus (RSV) and human metapneumovirus.

## Chest CT

The chest CT was obtained in a mobile CT scan unit (Alliance Medical equipped with GE lightspeed 16 slice scanner) upon arrival at the ED. CT scans were performed in caudo-cranial scanning direction without intravenous contrast injection, at 120kVp and 50–210 mAs, depending on their weight, using 16 x 1.25 collimation, 0.5s rotation time reconstructed at 1.25 mm slices with 1.25 mm increment. Patients were instructed to hold their breath if clinically possible. The procedure was performed taking all measures into account to prevent contamination of patients and personnel, which included cleaning procedures and use of protective equipment by patients (if possible) and personnel. Images were reconstructed using a moderately soft reconstruction filter ("DETAIL") at mediastinal window and a sharp reconstruction filter ("LUNG") at lung window settings. either suspicious or not suspicious for COVID-19 related pneumonia. The chest CT scans were read using a standardizing reporting scheme using the items reported as typical or atypical for COVID-19 [14, 24]. Scans were reported suspicious if findings reported as typical for COVID-19 were noted or non-suspicious if they were normal or showed findings typical for a bacterial pneumonia or respiratory bronchiolitis. Cases showing features of any viral infection, i.e. without the typical subpleural distribution of COVID-19, which is not usually seen in other viral pneumonia, were reported equivocal, but considered positive. Initial judgement of the CT scan was performed by a senior resident. The final reading and reporting were performed by an experienced chest radiologist within 12 hours of scanning. Both judgements were made with the specific request to check for signs of COVID-19 and, in many cases, included the nature and duration of the symptoms. Since the PCR results were available after 12–24 hours, both readers were unaware of the PCR test results.

### Data collection

From electronic medical records, we retrieved the following data: demographic data (age, sex); comorbidity; duration and severity of current disease; PCR results and other microbiological data; CT scan reports; discharge diagnosis. Severity of disease was classified in two ways: 1) using the severity score for community acquired pneumonia, quantified by the CURB-65 score (Confusion, Urea, Respiration, Blood pressure, Age; low/medium risk: 0–2 vs. high risk: 3–5 [25]), and 2) by establishing the absence or presence of sepsis using the SOFA score (Sepsis-related Organ Failure Assessment; 0–1 vs. ≥2 [26]).

### Data analysis and statistics

All statistical analyses were performed using IBM SPSS version 26 (Chicago, Illinois, USA) and MedCalc (https://www.medcalc.org/). We performed a descriptive analysis of baseline characteristics of included patients. Continuous variables were reported as medians with interquartile ranges (IQRs) and categorical variables as proportions. In case of missing values, valid percentages were used.

We compared the CT scan results with the RT-PCR testing results. Diagnostic accuracy of the CT scan in terms of sensitivity, specificity, positive and negative predictive value (PPV and NPV, resp.) and likelihood ratios (LRs; all with 95% confidence intervals (CI)) was assessed. Next, we calculated the diagnostic accuracy (sensitivity, specificity, PPV, NPV and LR) with respect to the severity of disease (CURB-65 score: low/medium vs. high risk) and the absence/ presence of sepsis (SOFA score: 0–1 vs. ≥2). The Chi Square test was used to compare the sensitivity and specificity between patients with low and high severity of disease.

A sample size calculation with regard to the diagnostic accuracy analysis was made based on the assumption of a prevalence of disease of 40%, and in order to detect a change in sensitivity from 70 to 90% with a power of 80% and a significance level of <0.05. The required sample size would be at least 78 [27].

Discordances between CT results and PCR testing results were further investigated by retrieving data on alternative diagnoses and, if possible, duration of symptoms (using ancillary viral/bacterial test results and discharge diagnoses in medical charts (e.g. pneumonia caused by influenza)). These data were analysed in a descriptive way.

## Results

### Patient characteristics

During the study period, 193 symptomatic ED patients had both a chest CT and RT-PCR test for diagnosing COVID-19. The median age of these patients was 66 years and 58.5% were male (Table 1). Most patients (72.5%) had one or more comorbidities. Cardiovascular comorbidity was highly prevalent, as was chronic pulmonary disease (19.7%). In total, 118 (61.1%) patients were admitted to the hospital.

Of the chest CT scans, 109 (56.5%) were judged as suspicious for COVID-19. In 83 (43.0%) patients, the PCR was positive for SARS-CoV-2 (Table 1). In 7 of these patients, a PCR was necessary on a second sample to confirm the suspected diagnosis, because the first PCR was negative or inconclusive.

### Diagnostic accuracy of chest CT for COVID-19 in all patients and subgroups

In 83 patients with a positive PCR, 74 CT scans were judged as suspicious for COVID-19 (Table 2). In all 110 patients with a negative PCR, 75 CT scans were judged as not suspicious for COVID-19. This results in a sensitivity of 89.2% (95%CI: 80.4–94.9%) and specificity of

**Table 1. Patient characteristics.**

| n (%), median (IQR) | All patients N = 193 |
|---|:---:|
| **Demographics** | |
| Age (years) | 66 (55–76) |
| Male | 113 (58.5) |
| **Comorbidity** | |
| No comorbidity | 53 (27.5) |
| Hypertension | 79 (40.9) |
| Diabetes mellitus | 31 (16.1) |
| Myocardial infarction | 31 (16.1) |
| Cerebrovascular disease | 23 (11.9) |
| Heart failure | 15 (7.8) |
| Peripheral vascular disease | 15 (7.8) |
| Chronic pulmonary disease | 38 (19.7) |
| Malignancy | 27 (14.0) |
| Chronic kidney disease | 17 (8.8) |
| Liver disease | 10 (5.2) |
| **Disease severity** | |
| CURB-65 score | 1 (0–2) |
| SOFA | 3 (1–4) |
| **Chest CT** | |
| CT suspicious for COVID-19 | 109 (56.5) |
| **RT-PCR** | |
| PCR SARS-CoV2 positive | 83 (43.0) |
| **Admission** | 118 (61.1) |

CURB-65: Confusion, Urea, Respiration, Blood pressure, Age; SOFA: Sepsis-related Organ Failure Assessment

68.2% (95%CI: 58.6–76.7%) of the CT for diagnosing COVID-19 (Table 3). The PPV was 67.9% (95%CI: 61.4–73.7%) and NPV 89.3% (95%CI: 81.6–94.0%).

## Diagnostic accuracy of the chest CT in relation to disease severity

Of all patients, 91.2% were classified as mild/medium risk pneumonia according to the CURB-65 score, with 71.0% having sepsis (SOFA score $\geq 2$, Table 3). The sensitivity of the CT tended to be higher (100.0%) in those with severe risk pneumonia than in patients with low/medium risk pneumonia (88.3%, p = 0.38). In patients with sepsis, sensitivity was significantly higher than in those without sepsis (95.5 vs. 62.5%, p<0.001).

## Analysis of discordant CT and RT-PCR results

In 44 (22.8%) patients discordant findings between CT and PCR were observed. In most cases, the CT scan was considered suspicious for COVID-19, while the PCR was negative (35/110,

**Table 2. Overview of chest CT and RT-PCR results.**

| | PCR SARS-CoV-2 positive | PCR SARS-CoV-2 negative | Total |
|---|:---:|:---:|:---:|
| **CT suspicious for COVID-19** | 74 | 35 | 109 |
| **CT not suspicious for COVID-19** | 9 | 75 | 84 |
| **Total** | 83 | 110 | 193 |

**Table 3. Diagnostic accuracy of Chest CT for diagnosing COVID-19 in all patients and in relation to disease severity.**

| | N (%) | Sensitivity %(95%CI) | Specificity %(95%CI) | PPV %(95%CI) | NPV %(95%CI) | LR+ (95%CI) | LR-(95%CI) |
|---|---|---|---|---|---|---|---|
| **Total group** | 193 (100) | 89.2 (80.4–94.9) | 68.2 (58.6–76.7) | 67.9 (61.4–73.7) | 89.3 (81.6–94.0) | 2.81 (2.11–3.72) | 0.16 (0.08–0.30) |
| **Disease severity** | | | | | | | |
| CURB-65 0–2 | 176 (91.2) | 88.3 (79.0–94.5) | 69.7 (59.7–78.5) | 69.4 (62.5–75.6) | 88.5 (80.4–93.5) | 2.91 (2.1–4.0) | 0.17 (0.09–0.31) |
| CURB-65 ≥3 | 17 (8.8) | 100.0 (54.1–100.0) | 54.5 (23.4–83.3) | 54.5 (38.6–69.6) | 100.0 | 2.20 (1.15–4.20) | 0.0 |
| SOFA score 0–1 | 56 (29.0) | 62.5* (35.4–84.8) | 70.0 (53.5–83.4) | 45.5 (31.2–60.5) | 82.4 (70.6–90.1) | 2.08 (1.14–3.82) | 0.54 (0.28–1.04) |
| SOFA score ≥2 | 137 (71.0) | 95.5* (87.4–99.1) | 67.1 (54.9–77.9) | 73.6 (66.5–79.6) | 94.0 (83.7–98.0) | 2.91 (2.07–4.08) | 0.07 (0.02–0.20) |

**95%CI**: 95% Confidence Interval; CURB-65: Confusion, Urea, Respiration, Blood pressure, Age; LR+, positive likelihood ratio; LR-, negative likelihood ratio; PPV, positive predictive value, NPV, negative predictive value; SOFA, Sepsis-related Organ Failure Assessment.

*significantly different with a p-value <0.001 (Chi-Square test)

31.8%). In the majority of these, the diagnosis at discharge was pulmonary infection (n = 26; 74.3%). In 9 of these 26 patients, other viral pathogens (Influenza A virus: n = 2; Human metapneumovirus: n = 5; Rhinovirus: n = 1, non-COVID corona virus: n = 1) and in 2, bacterial pathogens were found. In an additional 8 patients with pulmonary infection, COVID-19 disease was found unlikely based on the discharge diagnosis (all had at least 2 negative PCRs). However, in 7 patients with pulmonary infection, COVID-19 disease was found likely or could not be excluded. Median duration of symptoms at the moment of CT scanning and PCR was 5 days (in 1 less than 48 hours) in these 7 patients.

In 9 patients with a suspicious CT scan and a negative PCR, another diagnosis than pulmonary infection was made. Four patients had another pulmonary diagnosis (bronchiectasis (n = 1), asthma (n = 2), pleural effusion due to ascites (n = 1)), while in 4 patients, a cardiac diagnosis (heart failure (n = 3), acute coronary syndrome (n = 1)) was made.

In 9 patients (10.8%), CT scans were not suspicious for COVID-19, while the PCR was positive. In 2 of these patients, a second PCR was positive after a first negative test. In these cases, the CT scan was not repeated to check for new abnormalities. In one patient, respiratory symptoms were present for less than 48 hours, in 6 patients, symptoms were present for more than 48 hours, and in 2 patients, symptom duration was not clear.

## Discussion

In this prospective study in patients presenting at the ED with a clinical suspicion of COVID-19, we found that chest CT scan when compared to RT-PCR had a high sensitivity of 89% and an LR- of 0.16. Specificity was moderate (68%), with an LR+ of 2.81. Sensitivity of the CT was higher in patients with more severe disease—high CURB-65 score (≥3) or sepsis (SOFA ≥2) —than in those who were less severely ill. In 77% of all patients, the results of the chest CT and the PCR test were concordant, however in 11% of patients with a positive PCR, CT scans were not considered suspicious for COVID-19. In addition, in about a third of patients tested negative by PCR, the CT was positive. Most of these patients had a discharge diagnosis of pneumonia (74%), which was caused by another viral pathogen in one fourth of patients, while in 20%, COVID-19 could not be excluded.

The diagnostic ability of chest CT was found to be rather high, and in 77%, concordant findings of CT and PCR were found. Nevertheless, it should be noted that 11% of COVID-19 were missed and 32% were incorrectly assigned a suspected COVID-19. In a situation where isolating each patient separately is not possible, these patients are placed at a COVID-19 cohort, where they can be exposed to COVID-19 positive patients with possible devastating consequences.

The results of diagnostic accuracy should be interpreted with care for several reasons. One of these reasons is because it is known that both CT and PCR can be false negative in early stages of COVID-19 [5, 7, 16]. This is why we included repeated PCRs and explored discordant test results, which showed that in about 26% of patients with false positive CT scans, other viral pathogens were detected. Another reason is that this study was performed at the end of the respiratory virus season, and the added value of the CT scan might be lower during typical respiratory virus season. It can be argued, in addition, that with changing incidence of COVID-19, and thus, a changing pre-test probability, the added value of the CT scan will change. In other studies, the sensitivity of chest CT for COVID-19 was found to be higher, up to 97% in a study with 1014 Chinese patients, but at the cost of low specificity (25%) [14]. In another study in 81 Chinese patients with PCR proven COVID-19, sensitivity was 93% [10]. A lower sensitivity of 80% was reported in 30 symptomatic cruise ship passengers [17]. In contrast, PCR was found to be negative in 71% of 51 Chinese patients, of whom 98% had CT abnormalities [15]. We have chosen to consider all chest CTs with abnormalities that could be related to COVID-19 as suspicious, even though the changes were judged to compatible with non-COVID-19 related pneumonia or other respiratory conditions. However, in case no signs of a viral interstitial pneumonia were present, the CT scan was considered negative for COVID-19. Considering these cases negative would have improved the specificity, but at the cost of lower sensitivity. It must further be noted that the CT scans were judged with the specific request to search for COVID-19. This may have biased the radiologists into considering (equivocal) changes as COVID-19 related, thereby increasing sensitivity and decreasing specificity. Last, as COVID-19 is a new disease, and the first patient with COVID-19 presented in our hospital the 10[th] of March, it is possible that with increasing experience the accuracy of the judgment of the chest CT will improve over time.

Evaluation of discharge diagnosis revealed that in 7 (20%) of discordant, "false positive" CT scans, COVID-19 was found likely by the clinicians. Correction for these discordances would increase both sensitivity and specificity of the CT scan but this should be considered with care because of the small sample. Of the patients with false negative chest CTs (11%), only one patient had a documented short duration of symptoms (<48 hours). In total, 7 second PCRs were needed to diagnose COVID-19, which underlines the importance of good sampling and repeated testing to definitely exclude COVID-19 in symptomatic patients.

Not surprisingly, the sensitivity of the CT scan was higher in the more severely ill patients. In 17 patients with high-risk CURB-65 scores (≥3) and in 137 patients (71%) with sepsis (SOFA ≥2), sensitivity was 100% and 96%, resp. This finding was reported in other studies as well [2, 28] and is not surprising, because in more severe disease, more abnormalities can be expected to be found on the chest CT [7]. The high sensitivity for patients with a high-risk CURB-65 score was, however, at the cost of low specificity, but this was not the case for patients with sepsis. In addition, the numbers of patients in this subanalysis was small and confidence intervals were wide, so no hard conclusions on this topic can be drawn.

Interestingly, most patients (91%) presented with low/medium risk pneumonia (CURB-65 0–1). It is important to realize that 5 patients who were immediately intubated upon arrival at the ED received no CT scanning and were therefore not included in our study. However, although most included patients had no high risk pneumonia, many (71.0%) were septic.

The main advantage of the CT scan is that its' results are available almost immediately after scanning, in contrast to the PCR test, which may take up to 24 hours, although this usually takes less time. This advantage is dependent on the availability of a CT scan, personnel and a logistically well designed way to perform these scans. We rented an extra (mobile) CT scanner and placed it in a tent in front of the ED. During this time-frame, highly suspicious patients are kept in isolation, awaiting the PCR results. A chest CT scan can help to differentiate those

with high risk (suspicious CT) and those with low risk of COVID-19 (non-suspicious CT). It would be interesting to investigate whether combining the results of chest CT with clinical characteristics would increase the discriminatory value, which is extremely important especially if patients have to be placed in cohorts.

Our study has some limitations. External validity may be limited in this study due to its single centre set-up. In addition, especially in patients with mild symptoms who were not admitted to the hospital, no second PCR-testing was done after an initial negative result. A third limitation is that in the 5 patients who were intubated directly upon arrival at the ED, no CT scan was made. However, many patients who were seriously ill were included in the study, as demonstrated by the high proportion of patients being septic.

In conclusion, the diagnostic accuracy of chest CT in symptomatic ED patients is good, but not good enough to safely diagnose or exclude COVID-19, especially when patients are placed in cohorts. However, CT can be used as a quick tool to categorize patients into "probably positive" and "probably negative" cohorts.

## Supporting information

**S1 Data.**
(SAV)

## Acknowledgments

We are grateful to J. Lamain, M. de Jong, B. Vermeire and A. Wagner for their help in acquiring data.

## Author Contributions

**Conceptualization:** Hester A. Gietema, Patricia M. Stassen.

**Data curation:** Noortje Zelis, J. Martijn Nobel, Lars J. G. Lambriks, Lieke B. van Alphen, Astrid M. L. Oude Lashof, Irene C. Nelissen.

**Formal analysis:** Noortje Zelis, Irene C. Nelissen, Patricia M. Stassen.

**Investigation:** Noortje Zelis, J. Martijn Nobel, Lars J. G. Lambriks, Lieke B. van Alphen, Astrid M. L. Oude Lashof, Joachim E. Wildberger.

**Methodology:** Hester A. Gietema, Noortje Zelis, Joachim E. Wildberger, Patricia M. Stassen.

**Supervision:** Patricia M. Stassen.

**Writing – original draft:** Hester A. Gietema, Noortje Zelis, Patricia M. Stassen.

**Writing – review & editing:** Hester A. Gietema, Noortje Zelis, J. Martijn Nobel, Lars J. G. Lambriks, Lieke B. van Alphen, Astrid M. L. Oude Lashof, Joachim E. Wildberger, Irene C. Nelissen, Patricia M. Stassen.

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
