## [Decision Letter · Decision Letter 0]

26 May 2020

PONE-D-20-13081

CT in relation to RT-PCR in diagnosing COVID-19 in the Netherlands: a prospective study

PLOS ONE

Dear Dr. Stassen,

Thank you for submitting your manuscript to PLOS ONE. After careful consideration, we feel that it has merit but does not fully meet PLOS ONE’s publication criteria as it currently stands. Therefore, we invite you to submit a revised version of the manuscript that addresses the points raised during the review process.

Both reviewers had several concerns, especially regarding definition and statistical analysis, Please effectively respond to their comments in your revision.

We look forward to receiving your revised manuscript.

Kind regards,

Yu Ru Kou, PhD

Academic Editor

PLOS ONE

Journal Requirements:

2. Thank you for stating that informed consent was waived. Please also include this information in your ethics statement in the online submission form.

Reviewers' comments:

Reviewer's Responses to Questions

**Comments to the Author**

1. Is the manuscript technically sound, and do the data support the conclusions?

Reviewer #1: Partly

Reviewer #2: Yes

2. Has the statistical analysis been performed appropriately and rigorously? 

Reviewer #1: Yes

Reviewer #2: No

3. Have the authors made all data underlying the findings in their manuscript fully available?

Reviewer #1: No

Reviewer #2: Yes

4. Is the manuscript presented in an intelligible fashion and written in standard English?

Reviewer #1: Yes

Reviewer #2: Yes

5. Review Comments to the Author

Reviewer #1: PONE-D-20-13081

The goal of this publication is to evaluate the accuracy of CT for the diagnosis of COVID-19, using RT-PCR as the gold standard.

Strengths:

The topic of diagnosing COVID-19 with CT has been much debated. The study has some strengths compared to earlier literature. It is a prospective study of consecutive adult patients through the ED of a large volume center.

193 patients in a high prevalence area (43% had COVID) were included. Sensitivity of 89.2% and specificity of 68.2% is reported.

Weaknesses:

While “typical” imaging findings of COVID-19 are discussed, limited information is provided about how CT was actually used in the study to assess for COVID in a binary fashion. The suggestion is that many findings were considered positive: “In case of pneumonia, in which COVID-19 was unlikely but could not be excluded, the scan was judged positive.” More details are needed about the threshold for a “positive” CT.

The issue of blinding needs to be addressed. It is stated that CT readers were not aware of PCR status. But what were they aware of? Did they not presentation history, likelihood of COVID versus alternative diagnoses, etc. Along these lines, details should be provided about patients with and without COVID to address the question of whether COVID status might correlate with other characteristics in this cohort.

The attempt to discuss disease severity is welcome, as a criticism of earlier publications on the same topic (some included in the discussion) is that CT is good at detecting pneumonia in patients with pneumonia and COVID. Unsurprisingly, all COVID patients with severe risk for pneumonia had CT findings (what patient with pneumonia does not have CT findings?), but the specificity is poor.

The potential role of CT as a screening test needs to be developed. The conclusion makes the following statement about the diagnostic accuracy of CT: “not good enough to safely diagnose or exclude COVID-19.” If CT can neither rule in nor rule out disease, what practical role can it play for screening or diagnosis?

The conclusion need to couch the results in the context of a high prevalence environment. Discussion of how CT might perform in the lower prevalence environments we are likely to see in coming months should be included.

Specific points:

1. The details about discordant cases are interesting, but it is hard to draw conclusions from them. For this reason, the sensitivity and specificity numbers should not be revised to account for these cases.

2. CT is fast, but cumbersome to perform in the time of COVID-19. This should be added to the discussion.

Reviewer #2: Appraised by a revised tool for the quality assessment of diagnostic accuracy studies (QUADAS2), your research is qualified. Due to some limitation of PCR assay, the accuracy of chest CT as a diagnostic tool for COVID-19 is an important issue during the progress of pandemic. Although there were some published reports in this topic, we still need more studies to clarify the certainty of evidences especially from the countries other than China. Therefore, it is worth to publish your research before accomplishing the answers of the below issues.

1.Did you calculate the power for your diagnostic study by statistical software based upon the prospective design of this research? If no or insufficient power, please mention it as one of the limitations of your study.

2.You did not introduce the software for statistical analysis. You should demonstrate the 95% confidence interval of sensitivity, specificity, likelihood ratio, and other results.

3.You performed some sensitivity analyses including CURB-65 score and SOFA score to distinguish the different diagnostic value in different severity of patients. Is it possible to perform more sensitivity analyses toward CT findings? You mentioned that “Chest CT can show characteristic findings including areas of ground-glass, with or without signs of reticulation (so called “crazy paving pattern”), consolidative pulmonary opacities in advanced stages and the “reverse halo” sign. Since peripheral areas of ground glass are a hallmark of early COVID-19, which can easily be missed at chest X-rays, CT scanning has an advantage over chest X-rays in the early stages of COVID-19.” Can you calculate the diagnostic values of above characteristic findings in CT?

4.You described the detailed methods very well. You mentioned that “ The chest CT was obtained in a mobile CT scan unit (Alliance Medical equipped with GE148 lightspeed 16 slice scanner) upon arrival at the ED.” Please introduce more details about how to avoid nosocomial contamination in your policies.

6. PLOS authors have the option to publish the peer review history of their article (what does this mean?). If published, this will include your full peer review and any attached files.

Reviewer #1: No

Reviewer #2: No

---

## [Author Response · Author response to Decision Letter 0]

3 Jun 2020

Reviewers' comments:

Reviewer's Responses to Questions

Comments to the Author

1. Is the manuscript technically sound, and do the data support the conclusions?

Reviewer #1: Partly

Reviewer #2: Yes

2. Has the statistical analysis been performed appropriately and rigorously? 

Reviewer #1: Yes

Reviewer #2: No

3. Have the authors made all data underlying the findings in their manuscript fully available?

Reviewer #1: No

Reviewer #2: Yes

4. Is the manuscript presented in an intelligible fashion and written in standard English?

Reviewer #1: Yes

Reviewer #2: Yes

5. Review Comments to the Author

Reviewer #1: PONE-D-20-13081

The goal of this publication is to evaluate the accuracy of CT for the diagnosis of COVID-19, using RT-PCR as the gold standard.

Strengths:

The topic of diagnosing COVID-19 with CT has been much debated. The study has some strengths compared to earlier literature. It is a prospective study of consecutive adult patients through the ED of a large volume center.

193 patients in a high prevalence area (43% had COVID) were included. Sensitivity of 89.2% and specificity of 68.2% is reported.

Weaknesses:

While “typical” imaging findings of COVID-19 are discussed, limited information is provided about how CT was actually used in the study to assess for COVID in a binary fashion. The suggestion is that many findings were considered positive: “In case of pneumonia, in which COVID-19 was unlikely but could not be excluded, the scan was judged positive.” More details are needed about the threshold for a “positive” CT.

First, we would like to thank you for your thorough review in these extraordinary circumstances. Please note that the scans were judged in first stage of the pandemic and limited experience and guidelines existed. The judgement is therefore reflective of real clinical practice. The comment you made concerns the equivocal scans. A small part of the scans showed signs that might have been compatible with COVID-19, but also of other pulmonary diseases, such as bacterial pneumonia or pre- existent conditions. We decided to categorize these scans as positive for COVID-19. 

We realize that this could have been made more clearly in our manuscript and replaced the following line:

“In case of pneumonia, in which COVID-19 was unlikely but could not be excluded, the scan was judged positive.”

by 

“In case the chest CT was equivocal, i.e. it showed features compatible with COVID-19 but also with alternative pulmonary diseases, the scan was judged as a positive CT scan if at least some ground glass areas, that may be due to a viral infection, were present. However, if CT features were considered typical of a lobar pneumonia or respiratory bronchiolitis, the CT scan was judged as negative.” (lines 160-165, Methods).

In addition, we made the following changes to the Discussion:

We have chosen to consider all chest CTs with abnormalities that could be related to COVID-19 as suspicious, even though the changes were judged to be more likely compatible with non-COVID-19 related pneumonia....(and added) ... or other respiratory conditions. However, in case no signs of a viral interstitial pneumonia were present, the CT scan was considered negative for COVID-19 (lines 303-305) 

The issue of blinding needs to be addressed. It is stated that CT readers were not aware of PCR status. But what were they aware of? Did they not presentation history, likelihood of COVID versus alternative diagnoses, etc. 

The radiologists were not blinded for the clinical condition of the patient. They were aware of the pandemic and were working in the mobile CT scan unit, which was used for making chest CTs to screen for COVID19 only. In addition, the chest CTs were ordered using preprinted CT application forms, which mentioned the phrase: “COVID-19 screening” and, in many cases, the nature and duration of the complaints. This comment is therefore very true, and we decided to add the following sentence to the Methods section.

..... The final reading and reporting were performed by an experienced chest radiologist within 12 hours of scanning. ....

“Both judgements of the CT scan were made with the specific request to check for signs of COVID-19 and, in many cases, included the nature and duration of the symptoms.” (lines 166-168)

... Since the PCR results were available after 12-24 hours, both readers were unaware of the PCR test results.

And added the following to the Discussion:

Considering these cases negative would have improved the specificity, but at the cost of lower sensitivity. 

.... “It must be noted that the CT scans were judged with the specific request to search for COVID-19. This may have biased the radiologists into considering (equivocal) changes as COVID-19 related, thereby increasing sensitivity and decreasing specificity.” ...... (lines 306-309) 

Along these lines, details should be provided about patients with and without COVID to address the question of whether COVID status might correlate with other characteristics in this cohort.

We are not sure what the reviewer means with this comment. The focus of the manuscript was on the diagnostic accuracy of chest CT and not on the characteristics of COVID and non COVID patients.

The attempt to discuss disease severity is welcome, as a criticism of earlier publications on the same topic (some included in the discussion) is that CT is good at detecting pneumonia in patients with pneumonia and COVID. Unsurprisingly, all COVID patients with severe risk for pneumonia had CT findings (what patient with pneumonia does not have CT findings?), but the specificity is poor.

It is true that sensitivity in severely ill patients was higher, and (a bit, for sepsis) at cost of specificity. The radiologists, however, did not judge all pneumonia cases automatically as COVID-19. We now realize that this is not made perfectly clear to the readers, and kindly refer to the changes made according to your earlier remarks (first comment). In addition, we added the following to the Discussion:

“The high sensitivity for patients with a high-risk CURB-65 score was, however, at the cost of low specificity, but this was not the case for patients with sepsis.“ (lines 327-329) 

The potential role of CT as a screening test needs to be developed. The conclusion makes the following statement about the diagnostic accuracy of CT: “not good enough to safely diagnose or exclude COVID-19.” If CT can neither rule in nor rule out disease, what practical role can it play for screening or diagnosis?

Agreed, this could have been made more clear.

We added the following to the Conclusion/Abstract:

..”However, CT can be used as a quick tool to categorize patients into “probably positive” and “probably negative” cohorts. “ (lines 73-74; 358-359)

The conclusion need to couch the results in the context of a high prevalence environment. Discussion of how CT might perform in the lower prevalence environments we are likely to see in coming months should be included.

In the Discussion we already mentioned the influence of increasing incidence on the added value of CT (higher PPV, lower NPV), but the same is true for decreasing incidence (lower PPV, higher NPV).

We changed that sentence into the following...

It can be argued, in addition, that with increasing changing incidence of COVID-19, and thus, a higher changing pre-test probability, the added value of the CT scan will decrease.”... (lines 294-296)

Specific points:

1. The details about discordant cases are interesting, but it is hard to draw conclusions from them. For this reason, the sensitivity and specificity numbers should not be revised to account for these cases.

We agree that this extra calculation accounts for a small proportion of our patients, and therefore, these results were only shortly mentioned in the Discussion. We rephrased the sentence to de-emphasize the statement.

“Correction for these discordances would increase both sensitivity and specificity of the CT scan but this should be considered with care because of the small sample.” (lines 315-318)

2. CT is fast, but cumbersome to perform in the time of COVID-19. This should be added to the discussion.

Agreed, the advantage of the CT scan depends on its availability.

We added the following to the Discussion:

“This advantage is dependent on the availability of a CT scan, personnel and a logistically well designed way to perform these scans. We rented an extra (mobile) CT scanner and placed it in a tent in front of the ED.”...

(lines 340-342)

Reviewer #2: Appraised by a revised tool for the quality assessment of diagnostic accuracy studies (QUADAS2), your research is qualified. Due to some limitation of PCR assay, the accuracy of chest CT as a diagnostic tool for COVID-19 is an important issue during the progress of pandemic. Although there were some published reports in this topic, we still need more studies to clarify the certainty of evidences especially from the countries other than China. Therefore, it is worth to publish your research before accomplishing the answers of the below issues.

1. Did you calculate the power for your diagnostic study by statistical software based upon the prospective design of this research? If no or insufficient power, please mention it as one of the limitations of your study.

First, we would like to thank you for your thorough review in these strange and probably very busy times.

We made a sample size calculation on the assumption of a prevalence of disease of 40%, and in order to detect a change in sensitivity from 70 to 90% with a power of 80% and a significance level of <0.05. The required sample size would be at least 78. Our study was therefore sufficiently powered.

We added the following to the statistics section:

“A sample size calculation with regard to the diagnostic accuracy analysis was made based on the assumption of a prevalence of disease of 40%, and in order to detect a change in sensitivity from 70 to 90% with a power of 80% and a significance level of <0.05. The required sample size would be at least 78.” (lines 194-197)

2. You did not introduce the software for statistical analysis. You should demonstrate the 95% confidence interval of sensitivity, specificity, likelihood ratio, and other results.

In response to your first comment: you are correct, a sentence on the statistical software program used is added to the statistics section.

“All statistical analyses were performed using IBM SPSS version 26 (Chicago, Illinois, USA) and MedCalc (https://www.medcalc.org/).”(lines 181-182)

In response to the second comment: these were added, see the changes made throughout the manuscript. 

3. You performed some sensitivity analyses including CURB-65 score and SOFA score to distinguish the different diagnostic value in different severity of patients. Is it possible to perform more sensitivity analyses toward CT findings? You mentioned that “Chest CT can show characteristic findings including areas of ground-glass, with or without signs of reticulation (so called “crazy paving pattern”), consolidative pulmonary opacities in advanced stages and the “reverse halo” sign. Since peripheral areas of ground glass are a hallmark of early COVID-19, which can easily be missed at chest X-rays, CT scanning has an advantage over chest X-rays in the early stages of COVID-19.” Can you calculate the diagnostic values of above characteristic findings in CT?

This analysis is unfortunately not possible to make. There are several signs suggestive of COVID-19 on chest CT. This study was not designed to investigate the diagnostic value of the individual signs and lacks power to do this in retrospect. 

4.You described the detailed methods very well. You mentioned that “ The chest CT was obtained in a mobile CT scan unit (Alliance Medical equipped with GE148 lightspeed 16 slice scanner) upon arrival at the ED.” Please introduce more details about how to avoid nosocomial contamination in your policies. 

The mobile CT scan unit was placed in a tent in front of the ED. The tent was divided into compartments with beds each being surrounded by walls. Personnel was fully protected by wearing personal protection material.

The CT scan was cleaned after each scan. 

We added this to the Methods section:

“The procedure was performed taking all measures into account to prevent contamination of patients and personnel, which included cleaning procedures and use of protective equipment by patients (if possible) and personnel.”

(lines 154-156)

---

## [Decision Letter · Decision Letter 1]

10 Jun 2020

PONE-D-20-13081R1

CT in relation to RT-PCR in diagnosing COVID-19 in the Netherlands: a prospective study

PLOS ONE

Dear Dr. Stassen,

Thank you for submitting your manuscript to PLOS ONE. After careful consideration, we feel that it has merit but does not fully meet PLOS ONE’s publication criteria as it currently stands. Therefore, we invite you to submit a revised version of the manuscript that addresses the points raised during the review process.

One reviewer still raised some important issues that need to be adequately addressed.

We look forward to receiving your revised manuscript.

Kind regards,

Yu Ru Kou, PhD

Academic Editor

PLOS ONE

Reviewers' comments:

Reviewer's Responses to Questions

**Comments to the Author**

1. If the authors have adequately addressed your comments raised in a previous round of review and you feel that this manuscript is now acceptable for publication, you may indicate that here to bypass the “Comments to the Author” section, enter your conflict of interest statement in the “Confidential to Editor” section, and submit your "Accept" recommendation.

Reviewer #1: (No Response)

Reviewer #2: All comments have been addressed

2. Is the manuscript technically sound, and do the data support the conclusions?

Reviewer #1: Partly

Reviewer #2: Yes

3. Has the statistical analysis been performed appropriately and rigorously? 

Reviewer #1: Yes

Reviewer #2: Yes

4. Have the authors made all data underlying the findings in their manuscript fully available?

Reviewer #1: Yes

Reviewer #2: Yes

5. Is the manuscript presented in an intelligible fashion and written in standard English?

Reviewer #1: Yes

Reviewer #2: Yes

6. Review Comments to the Author

Reviewer #1: (No Response)

Reviewer #2: The authors have adequately addressed the reviewers’ comments. The revised manuscript is qualified and acceptable for publication in the PLOS ONE.

7. PLOS authors have the option to publish the peer review history of their article (what does this mean?). If published, this will include your full peer review and any attached files.

Reviewer #1: No

Reviewer #2: No

---

## [Author Response · Author response to Decision Letter 1]

17 Jun 2020

Dear prof Kou and dear reviewer,

Thank you for your review. Below, we respond to the comments made.

Reviewer's Comments:

1.Modestly better with more disclaimers. Still this is a study in a high prevalence area where unspecified imaging criteria are used by non-blinded radiologists to call COVID-19 on CT. More disclaimers are needed, and overall the results are not of clear value moving forward.

We understand from your comments that the way the CT scans were judged can be made more clear. The judgement was made in the same way as in other studies.

We tried to clarify this by changing the text in the Methods section (lines 158-164).

...”The chest CT scans were read using a standardizing reporting scheme using the items reported as typical or atypical for COVID-19 (14, 24). Scans were reported suspicious if findings reported as typical for COVID-19 were noted or non-suspicious if they were normal or showed findings typical for a bacterial pneumonia or respiratory bronchiolitis. Cases showing features of any viral infection, i.e. without the typical subpleural distribution of COVID-19, which is not usually seen in other viral pneumonia, were reported equivocal, but considered positive.”...

We are not sure which additional disclaimers are needed. We think we addressed these disclaimers already. All studies on this topic to date, as far as we know, have been performed under similar circumstances. This study focused on the accuracy of the CT scan in real practice, in which clinical information is always available to radiologists, and in a time frame during which the prevalence of COVID-19 was high. 

We kindly refer to the lines 166-168 in the Methods section:

Both judgements of the CT scan were made with the specific request to check for signs of COVID-19 and, in many cases, included the nature and duration of the symptoms. Since the PCR results were available after 12-24 hours, both readers were unaware of the PCR test results.

and to the lines 309-312 in the Discussion:

It must further be noted that the CT scans were judged with the specific request to search for COVID-19. This may have biased the radiologists into considering (equivocal) changes as COVID-19 related, thereby increasing sensitivity and decreasing specificity.

And to the paragraph on the disclaimers (lines 291-314) in the Discussion in which we emphasize the circumstances that have to be taken into account when interpreting the results of chest CT.

Nevertheless, to make these disclaimers more clear, we changed the text of the Discussion: (lines 291-296)

...”The results of diagnostic accuracy should be interpreted with care for several reasons. One of these reasons is because it is known that both CT and PCR can be false negative in early stages of COVID-19 (5, 7, 16)....Another reason...”

With regard to your remark on moving forward:

We feel that investigating the accuracy of chest CT was correctly done in this study in contrast to earlier (in general much smaller; 81, 30, 51 patients resp. (ref 10,17, 15)) studies that sometimes included PCR positive patients only (Shi, ref 10) or mainly asymptomatic patients (shohei, ref 17). We included a casemix of positive and negative patients, who were assessed at the ED because they had respiratory symptoms and/or other symptoms suggestive of COVID-19. We further included an analysis of discordant results, which others did not.

2. Also, information about other patients characteristics would be helpful to determine if non-imaging criteria correlated with COVID status, and influenced the decision of radiologists. For example, did the COVID patients have fevers, and this prompted radiologists to call them positive on imaging.

Before scanning, the patients were triaged by a nurse, whose main task was to evaluate whether a CT scan was needed or not, following a protocol (eg. contact with COVID positive patient, respiratory symptoms). He or she filled out the CT request form, which was labelled as “COVID-19 screening”, and included questions on duration and nature of symptoms. Due to crowding, these data were not filled in systematically, and therefore these could not be retrieved in a reliable way. Not until the CT scan was made, the doctor assessed the patient.

We kindly refer to the Methods section, lines 165-167, in which we address this topic. 

Both judgements of the CT scan were made with the specific request to check for signs of COVID-19 and, in many cases, included the nature and duration of the symptoms.

With kind regards, on behalf of the authors,

Patricia Stassen

---

## [Decision Letter · Decision Letter 2]

24 Jun 2020

CT in relation to RT-PCR in diagnosing COVID-19 in the Netherlands: a prospective study

PONE-D-20-13081R2

Dear Dr. Stassen,

We’re pleased to inform you that your manuscript has been judged scientifically suitable for publication and will be formally accepted for publication once it meets all outstanding technical requirements.

Kind regards,

Yu Ru Kou, PhD

Academic Editor

PLOS ONE

Additional Editor Comments (optional):

Reviewers' comments:

Reviewer's Responses to Questions

**Comments to the Author**

1. If the authors have adequately addressed your comments raised in a previous round of review and you feel that this manuscript is now acceptable for publication, you may indicate that here to bypass the “Comments to the Author” section, enter your conflict of interest statement in the “Confidential to Editor” section, and submit your "Accept" recommendation.

Reviewer #1: All comments have been addressed

2. Is the manuscript technically sound, and do the data support the conclusions?

Reviewer #1: Partly

3. Has the statistical analysis been performed appropriately and rigorously? 

Reviewer #1: N/A

4. Have the authors made all data underlying the findings in their manuscript fully available?

Reviewer #1: Yes

5. Is the manuscript presented in an intelligible fashion and written in standard English?

Reviewer #1: Yes

6. Review Comments to the Author

Reviewer #1: defer to editors at this point.

7. PLOS authors have the option to publish the peer review history of their article (what does this mean?). If published, this will include your full peer review and any attached files.

Reviewer #1: No

---

## [Editor Report · Acceptance letter]

29 Jun 2020

PONE-D-20-13081R2 

CT in relation to RT-PCR in diagnosing COVID-19 in the Netherlands: a prospective study 

Dear Dr. Stassen:

I'm pleased to inform you that your manuscript has been deemed suitable for publication in PLOS ONE. Congratulations! Your manuscript is now with our production department. 

Kind regards, 

on behalf of

Dr. Yu Ru Kou 

Academic Editor

PLOS ONE